

# Understanding text-based persuasion and support tactics of concerned significant others

Katherine van Stolk-Cooke[1,2], Marie Hayes[1,2], Amit Baumel[2] and Frederick Muench[1,2]

[1] Research Foundation for Mental Hygiene, Inc., NY, United States
[2] Current affiliation: Department of Psychiatry, North Shore-Long Island Jewish Health System, Great Neck, NY, United States

## ABSTRACT

The behavior of concerned significant others (CSOs) can have a measurable impact on the health and wellness of individuals attempting to meet behavioral and health goals, and research is needed to better understand the attributes of text-based CSO language when encouraging target significant others (TSOs) to achieve those goals. In an effort to inform the development of interventions for CSOs, this study examined the language content of brief text-based messages generated by CSOs to motivate TSOs to achieve a behavioral goal. CSOs generated brief text-based messages for TSOs for three scenarios: (1) to help TSOs achieve the goal, (2) in the event that the TSO is struggling to meet the goal, and (3) in the event that the TSO has given up on meeting the goal. Results indicate that there was a significant relationship between the tone and compassion of messages generated by CSOs, the CSOs' perceptions of TSO motivation, and their expectation of a grateful or annoyed reaction by the TSO to their feedback or support. Results underscore the importance of attending to patterns in language when CSOs communicate with TSOs about goal achievement or failure, and how certain variables in the CSOs' perceptions of their TSOs affect these characteristics.

**Submitted** 29 May 2015
**Accepted** 13 July 2015
**Published** 18 August 2015

Corresponding author
Frederick Muench,
fmuench@nshs.edu

## INTRODUCTION

The past few decades have witnessed an increased focus on how concerned significant others (CSOs) can impact the health and wellness of individuals attempting to meet health goals (*Zimmerman & Connor, 1989*; *Franks, Campbell & Shields, 1992*; *Hurdle, 2001*; *Gallant, 2013*). Resources have been designed specifically for CSOs, ranging in scope from educational initiatives by the National Alliance on Mental Illness (NAMI) for any number of mental health issues to targeted support groups like Al Anon for CSOs of drinkers. In addition to these, behavioral interventions, such as Community Reinforcement Approach and Family Training (CRAFT), have been designed to optimize the way CSOs communicate with their target significant others (TSOs), with the goal of

engaging TSOs in appropriate treatments (*Meyers et al., 1998*). Particularly for CSOs with a substance abusing TSO, the CRAFT program has been shown to be quite successful in its treatment engagement aims (*Roozen, De Waart & Van Der Kroft, 2010*). In essence, the growing consensus among researchers and health care professionals is that TSOs require environments that are supportive of behavior change, and that effective communication between CSOs and TSOs is central to fostering meaningful change beyond the clinic.

The ways in which we communicate with the significant others in our lives are diversifying rapidly with the growing use of technology for social purposes, from emailing to posting on social media to text messaging via mobile phone. However, these media are also being harnessed to better understand human computer interactions and to promote health. In fact, research over the last 40 years reveals that individuals are more honest and forthcoming when communicating via a digital medium (*Greist et al., 1973*; *Lucas et al., 2014*), making the ways in which we communicate with our loved ones via technology an important area of study. Although there is a growing literature on using ubiquitous technologies (e.g., mobile phones) to target problem behaviors (*Free et al., 2013*), little is known about how behavior change-oriented communication conducted via technology impacts that behavior. Some research has examined how the tone and structure of a message can have an impact on receptivity and engagement in a technology-mediated interaction about behavior change, revealing that polite and gain-framed content improves message receptivity (*Bickmore et al., 2007*; *Muench et al., 2014*). These studies point to the importance of understanding technology-mediated communication styles between CSOs and TSOs in order to optimize positive communication strategies and outcomes. Examining social support and persuasive messaging tactics from the perspective of the CSO may shed further light on this subject.

## Social support and CSOs

For many decades, the role of social support in the enhancement of health and wellness has been a topic of interest to scientists and practitioners across many related disciplines, including medical, psychological and sociological lines of inquiry (*Caplan, 1979*; *Cohen & Syme, 1985*; *Tracy & Whittaker, 1990*; *DiMatteo, 2004*). Under the umbrella of social support, a large and comprehensive literature on supportive communication has emerged, spanning face-to-face and technology-mediated communication (*Burleson & MacGeorge, 2002*; *Adams, Baumer & Gay, 2014*). Some of the earliest work on social support offered definitions of this construct that tie it strongly to elements of compassion, such that the target of social support is made aware that (s)he belongs to a community and is loved, cared for and esteemed (*Moss, 1973*; *Cobb, 1976*). However, much of the literature on technology-mediated supportive communication has not specifically examined communication enacted within pre-established familial, romantic or platonic relationships, instead focusing on communication enacted via online support groups and communities tailored to the behavior or health issue of interest (*Braithwaite, Waldron & Finn, 1999*; *Adams, Baumer & Gay, 2014*). This is striking, since one might argue that social support in our everyday lives comes predominantly from our CSOs, who offer us

their support and feedback both when we request it and when we do not. What's more, few studies have sought to understand social support enacted by CSOs based on the goal that they would like their TSOs to meet. In effect, little is known about how the CSO's agenda for their TSO may impact their supportive communication style. Understanding the language styles of CSOs in brief technology-mediated communication settings like short text-based messaging can reveal important information about the mechanisms and nuances of behavior change promotion within the context of these relationships. This can help us build interventions for CSOs attempting to persuade, motivate and/or support their loved ones to improve their lives.

## Persuasive communication

Research on persuasion has historically focused on how various components of language (e.g., the degree to which a message is tailored to the individual) impact the persuasiveness of a message (*Noar, Harrington & Aldrich, 2009*; *Oinas-Kukkonen & Harjumaa, 2009*). Within the context of behavior change research and treatment, this has resulted in the development of a number of treatments with a list of motivational language do's and don'ts. Perhaps the most notable example of this phenomenon is the development and dissemination of Motivational Interviewing as a clinical technique to promote behavior change in individuals by targeting ambivalence and promoting change talk through reflective listening and persuasive techniques (*Miller & Rollnick, 1991*).

Beyond therapeutic interventions, social psychological research on person-to-person communication reveals that individuals often plan how they will communicate based on their expectations of the individual attributes, interpersonal relationship and environmental context of the person with whom they intend to speak (*Burgoon, 1993*). These expectations can, in turn, impact the outcome of the communication. Such research suggests that language is a rules-based system in which people develop a pattern of anticipated norms with regard to language usage in any given situation (*Burgoon & Miller, 1985*). This language expectancy construct is particularly relevant to close relationships, in which individuals develop habitual self-reinforcing communication patterns. Therefore, understanding how the content of brief persuasive communication in different behavior change scenarios may relate to CSOs' expectations of their TSOs is an important step in developing communication tools for CSOs in the digital age.

## The current study

This study examined the language content and other characteristics of brief text-based messages generated by CSOs to motivate their TSOs to achieve a behavioral goal of the CSO's choosing. CSOs were asked to describe the goal they wanted their TSO to achieve, and to generate brief messages to their TSO for three behavior change scenarios: (1) to help them achieve the goal, (2) in the event that the TSO is struggling to meet the goal, and (3) in the event that the TSO has given up on meeting the goal altogether. Messages were then analyzed via linguistic analysis software and coded across a number of variables pertaining to language content and semantics. The aim of the study was to determine whether any global characteristics in behavior change messages existed based on the CSO-TSO

relationship and the three scenarios (above) for which each message was generated, and how variables in the CSOs' perceptions of their TSOs, such as motivation and expectations of a specific CSO reaction, might moderate message characteristics such as tone and compassion in an effort to inform the development of interventions for CSOs and TSOs.

## METHOD

### Recruitment

This study was approved by the New York State Psychiatric Institute Institutional Review Board (NYSPI IRB: #6625) and was part of the pilot intervention development work for a mobile adaptive alcohol intervention. Participants were recruited online through Amazon.com, Inc.'s online labor market, Amazon Mechanical Turk (MTurk). MTurk is a communication platform through which *workers* can be contracted to perform tasks that require human intelligence (e.g., consumer surveys or beta testing) in exchange for compensation by the *requesters* who publish the tasks. These tasks—called *human intelligence tasks* (HITs)—can range from one brief question to a 30-minute survey. Over the last few years, MTurk has been used for social sciences research with results similar to other sampling methods when certain validity checks were included in the design (*Mason & Suri, 2012*).

### Study eligibility

MTurk worker qualifications for this study included a HIT approval rate of 95% or greater out of at least 500 completed HITs. This ensured a sample of workers whose work on previous HITs had been consistently deemed acceptable by other requesters, as well as a sample who demonstrated an appropriate degree of computer and Internet literacy. The subject pool was further limited to participants who were located in the United States. Workers who met these qualifications could view our HIT, which was labeled with the following description: *This brief survey will ask you questions about helping friends and family improve their lives*, and published through our requester account, *Columbia University Research*. Eligible workers could follow a Web link to an external, Web-based survey hosted by Survey Monkey, which has been used as a survey host in numerous research studies. Prior to completing the survey, participants completed a brief consent form for anonymous survey-based research, which also provided investigator and IRB contact information. In the consent form, participants were informed that the study's aim was to better understand peoples' personal goals and how people support other individuals who are attempting to change a behavior. Once participants completed the survey, they were provided with a survey code to enter into their MTurk account to await requester review and compensation in the amount of $0.50.

For the purposes of maintaining anonymity, we could not link the survey to the participants' MTurk accounts, but included several a priori validity checks for anonymous survey research in both the survey and our MTurk requester account. These validity checks were included in accordance with the Checklist for Reporting Results of Internet E-Surveys (CHERRIES; *Eysenbach, 2004*). Our safeguards included blocking IP addresses once the

survey was opened by a worker in order to bar them from retaking it, and omitting responses of users who did not type cogent responses to open-ended questions and/or gave conflicting answers to a duplicated message preference question.

Although we were unable to match an individual worker to his or her survey responses, we were able to view the total amount of time each worker spent on the HIT in MTurk. As our survey should take a minimum of 5 min to complete, we rejected the work of participants who spent fewer than 5 min completing it.

## Participants

In total, 115 participants took the survey. Of those, 96 were included in the final sample: all 19 of those who were excluded were excluded either because they did not complete the survey in its entirety or because the content they contributed was illegible or did not pertain to the questions they were asked.

## Assessments

The assessment contained approximately 16 items, which were presented in groups of approximately 6 items per screen (See Appendix S1).

Participants were asked to identify their relationship to a loved one (e.g., parent, spouse, friend) who they believed should change a behavior in order to improve his or her life, and to identify the cognitive or behavioral goal they wanted their loved one to achieve. For the intents and purposes of this paper, we will refer to participants as Concerned Significant Others (CSOs) and to their loved ones as Target Significant Others (TSOs). We chose to leave the definition of the term "goal" open-ended, allowing the CSO to choose any goal that came to mind. These goals did not have to be health related. CSOs were told that the TSO's target goal could be anything from exercising more to being more assertive to cutting down on drinking, and that there were no wrong goals (see Appendix S1).

CSOs were then asked to generate the following: (1) a brief, text-based message to help the TSO meet the generated goal, (2) a brief, text-based message in the event that the TSO was not meeting the goal, and (3) a brief, text-based message in the event that the TSO had completely given up on meeting the goal. Once CSOs generated these three messages, they were provided with a series of process rulers that have been used in previous research (*Sobell & Sobell, 1996*) relating to the generated goal, including goal salience, severity of the possible consequences of failing to meet the goal, and goal efficacy. We assessed CSO perceptions of their relationship to their TSO and their perceptions of the TSO change process using five process rulers pertaining to the CSO's (1) perception of the closeness of their relationship to the TSO, (2) perception of their TSO's motivation to meet their goal, (3) belief that their TSO would take their advice were they to offer it, (4) perception of the severity of consequences if their TSO were to fail to meet their goal, and (5) frustration with the TSO's current progress (or lack thereof) toward meeting their goal. CSOs were additionally asked a multiple choice question about the type of reaction that they would expect from their TSO were they to offer suggestions or support to meet the goal. Finally, CSOs were asked to provide basic demographic information including their age, gender, race and ethnicity.

## Basic linguistic analysis

To assess basic linguistic features of content generated by CSOs, we conducted an exploratory linguistic analysis on all goals and messages generated using the 2007 version of the Linguistic Inquiry and Word Count (LIWC) software developed by *Pennebaker, Francis & Booth (2001)*. This software categorizes words across a variety of language dimensions, including basic descriptors (e.g., total word count, average number of words per sentence) standard linguistic processes (e.g., number of pronouns, articles and verbs used), and psychological processes (e.g., social, affective, cognitive, perceptual and biological processes).

## Message coding

TSO goals were coded into three broad categories based on their subject matter: physical health and well-being, competence and mastery, and personal fulfillment. Goals within these categories were then subcoded into more specific groupings. We have used these coding variables in previous research on the content of text-based messages for behavior change (*Muench et al., 2014*).

We developed a number of coding schemas to assess three semantic characteristics of particular interest in the context of the CSO-TSO relationship, including: (1) positive, negative, or neutral tone, (2) presence or absence of a compassionate tone, and (3) gain- or loss-framed orientation of messages. Table 1 presents this coding schema, along with an example message for each coding variable.

We also coded all messages generated by CSO's based on a schema developed by *Cutrona & Suhr (1994)* to classify different types of supportive messages. Although variations on this schema and the definitions of its variables have emerged over time (*Adams, Baumer & Gay, 2014*), we used the original schema and definitions as follows: (1) *emotional support*, defined as an expression of care, concern and/or sympathy; (2) *esteem support*, defined as reassurance of worth, an expression of liking for or confidence in the TSO; (3) *network support*, defined as an expression of connection or belonging in a social community; (4) *informational support*, defined as information and/or advice; and (5) *tangible assistance*, defined as an offer of money, physical intervention or other material aid. We added an additional variable to this schema to categorize messages that were entirely unsupportive.

Two researchers independently coded all text messages using the taxonomy described above. Each was given a definition and example of a message for each category. Interrater agreement when messages from all three messaging scenarios were merged revealed 89.6% agreement on tone, 78.1% on compassion, and 83.7% on gain vs. loss-framing, and 94% on social support type. In cases where there was disagreement between coders about the categorization of a message (e.g., 11.4% of tone messages), the message was discussed and subsequently placed either in a category based on agreement between both coders or marked as uncoded and excluded from analyses. Of the 288 messages provided by the participants, 5 (1.7%) messages were left uncoded within the tone category, 4 (1.4%) were left uncoded within the compassion category, 4 (1.4%) were left uncoded in the

**Table 1  CSO message coding.**

| Coding category | Coding variable | Example message |
|---|---|---|
| Tone | Positive<br>*Message is affirmative, encouraging, or favorable towards the TSO and goal achievement.* | "Hey Dad, it looks like you've lost some weight, keep at it!" |
| | Negative<br>*Message is pessimistic, critical, or dismissive of the TSO and his/her behavior change attempts.* | "So long and good luck. Don't come crawling back." |
| | Ambiguous<br>*Message can be read as either positive or negative in tone, depending on the specific context/relationship of CSO and TSO; message in itself doesn't strongly confirm either positive or negative tone.* | "Call your children more often." |
| Compassion | Compassionate<br>*Message conveys a tone of care and concern for the TSO, or encourages the TSO to be self-compassionate.* | "I care a lot about you and I know you can do this!" |
| | Not compassionate<br>*Message contains no indication of CSO's care or concern for the TSO.* | "You smoke too much. You should stop." |
| Gain vs. loss orientation | Gain-framed<br>*Message emphasizes the positive results of goal achievement.* | "The harder you try, the more money you'll have!" |
| | Loss-framed<br>*Message emphasizes the negative consequences of goal failure.* | "It's your life, I know, but your weight is literally killing you." |
| | Neither<br>*Message refers to neither positives of goal achievement nor consequences of goal failure.* | "Don't get discouraged." |

gain- vs. loss-framing category and 3 (1.1%) in the social support category due to rater disagreement about the proper classification of the message.

## Data analysis

Since this was an exploratory study, we provide descriptive statistics for each rating. We conducted Z-tests to see if there were significant differences in the proportions of linguistic content, tone, compassion, or social supportiveness of messages based on the messaging scenario (e.g., in the event TSO is not meeting goal) for which they were written. Simple bivariate correlations, chi-Square and ANOVAs were conducted to assess the relationship between the tone or compassion of a message and the CSO's expectation of a TSO reaction to support or feedback and the CSO's perceptions of the TSO relationship and change process (e.g., the TSO's motivation to meet their goal). Multiple linear and logistic regressions were performed to assess the unique variance for all significant associations.

## RESULTS

### Overview

Demographics are presented in Table 2. The sample assessed was predominantly young, white, and male: 62 out of 96 participants (64.6%) were between the ages of 18 and 30,

**Table 2 Demographics.**

| Variable | | n (%) |
|---|---|---|
| Age (years) | 18–30 | 62 (64.6) |
| | 31–40 | 21 (21.9) |
| | 41-older | 13 (13.5) |
| Gender (%female) | | 28 (29.5) |
| Race | Black | 3 (3.1) |
| | White | 83 (86.5) |
| | Asian | 7 (7.3) |
| | Other | 3 (3.1) |
| Ethnicity | Hispanic | 4 (4.2) |
| Relationship to TSO | Spouse/partner | 25 (26.0) |
| | Parent | 27 (28.2) |
| | Sibling | 20 (20.8) |
| | Close Friend | 20 (20.8) |
| | Other | 4 (4.1) |
| TSO goal type | Physical health & wellbeing | 51 (53.1) |
| | Competence & mastery | 32 (33.3) |
| | Personal fulfillment | 13 (13.5) |

83 out of 96 (86.5%) were white, and 68 out of 96 (70.5%) were male. CSOs primarily identified family members as their TSO of choice, with 25 out of 96 participants (26.0%) choosing to target their spouse, 27 out of 96 (28.2%) choosing to target a parent, and another 20 out of 96 (20.8%) choosing to target a sibling. The vast majority of remaining participants—20 out of 96 (20.8%)—chose to target a close friend.

CSOs generated a variety of goals for their TSOs. In total, 51 out of 96 participants (53.1%) generated goals related to their TSOs' physical health and/or wellbeing (e.g., "I would like him to lose weight"). Within this group, 25 out of 51 participants (49.0%) generated goals for their TSOs pertaining explicitly to smoking, drinking, or other substance use cessation. 32 out of 96 participants (33.3%) generated goals related to competence and mastery (e.g., "I want her to finish her Ph.D. dissertation"). Finally, 13 out of 96 participants (13.5%) generated goals related to personal fulfillment (e.g., "I want her to be happy").

## CSO perceptions of TSO motivation & expectation of a reaction

As shown in Table 3, the most common reaction CSOs expected from their TSOs in response to CSO messages was gratitude, followed by annoyance or irritation. The least common reactions that CSOs expected were anxiety or panic, followed by hurt. Based on mean scores, most CSOs reported that their relationships with their TSOs were close ($M = 7.65, SD = 2.11$). However, CSOs reported being generally frustrated with the efforts of their TSOs to meet their goal ($M = 6.36, SD = 2.59$) and low perceived motivation on the part of TSOs to meet the goal ($M = 5.19, SD = 2.49$), despite the CSOs' fears of severe negative consequences of TSO inaction ($M = 6.55, SD = 2.39$).

**Table 3  TSO-CSO relationship & goal achievement rulers ($n = 96$).**

| Variable | | Percent | Mean (SD) |
|---|---|---|---|
| Expected reaction to CSO suggestions or support | Gratitude | 52.1 | |
| | Annoyance | 34.4 | |
| | Excitement | 21.9 | |
| | Embarrassment | 20.8 | |
| | Detachment | 15.6 | |
| | Anger | 14.6 | |
| | Hurt | 12.5 | |
| | Anxiety | 7.3 | |
| TSO-CSO closeness[a] | | | 7.65 (2.11) |
| TSO motivation[b] | | | 5.19 (2.49) |
| Will TSO take CSO advice[c] | | | 5.62 (2.25) |
| Consequence severity of goal failure[d] | | | 6.55 (2.39) |
| CSO frustration[e] | | | 6.36 (2.59) |

**Notes.**

[a] 1, estranged; 10, extremely close.
[b] 1, not at all motivated; 10, extremely motivated.
[c] 1, TSO will do the opposite of CSOs suggestion; 10, TSO will strive to take advice.
[d] 1, not at all severe; 10, extremely severe.
[e] 1, not at all frustrated; 10, extremely frustrated.

## LIWC analysis by messaging scenario

The total word count for messages generated for each messaging scenario were as follows: 1,356 words generated for messages to help TSOs meet their goal, 1,307 words generated in messages for the event in which the TSO was struggling to meet their goal, and 1,390 words generated on messages for the event in which the TSO had given up on meeting their goal. The average number of words per sentence was 12.56 in the first scenario, 12.33 in the second scenario, and 10.45 in the third scenario.

### Linguistic processes.

Use of negations (e.g., no, not, don't, never) increased significantly from the first to the third messaging scenario ($Z = -3.82$, $P < .001$, two-tailed). Use of punctuation also increased significantly from the first to the third messaging scenario ($Z = -2.73$, $P < .01$, two-tailed).

### Psychological processes

No significant differences in psychological processes existed across scenarios. However, within the affective process category, the number of words identified by the system as pertaining to positive emotion declined significantly from the first to the third messaging scenario ($Z = 2.05$, $P < .05$, two-tailed). No significant difference was found between the number of words identified by the system as pertaining to negative emotion between scenarios.

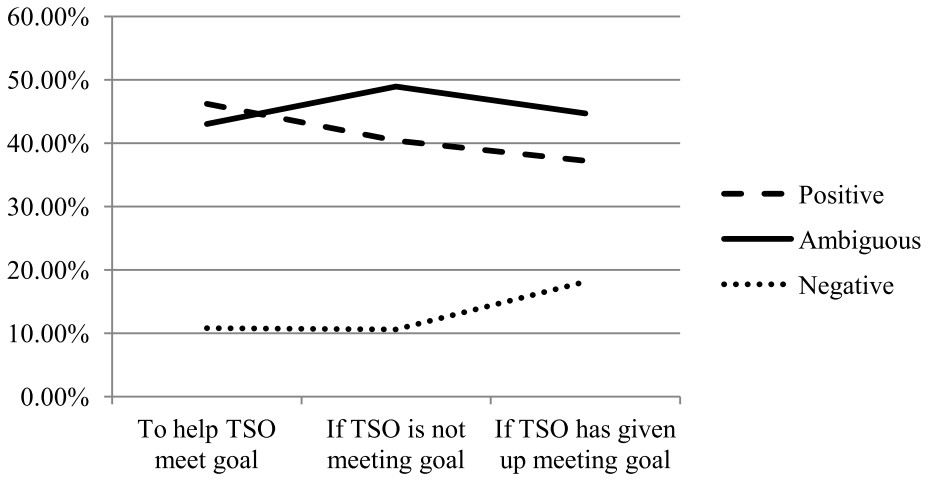

**Figure 1** **Tone by messaging scenario.**

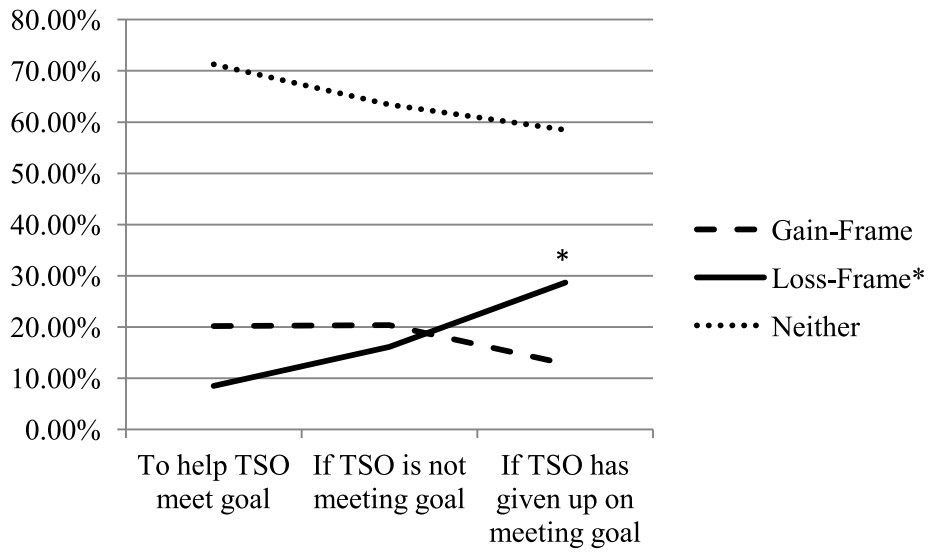

*Significant

**Figure 2** **Gain- and loss-framing by messaging scenario.**

## Message content by messaging scenario

None of the changes in the tone of messages between the three messaging scenarios were found to be significant (see Fig. 1), though there is a trend for positive messages to decrease and negative messages to increase in the event that the TSO has given up their goal.

The presence or absence of compassionate content in messages stayed virtually constant between messaging scenarios, varying by no more than a single participant in each. Overall, 33 out of 96 participants (35.1%) included compassionate content in their messages both to help their TSO meet their goal and in the event that the TSO was struggling to meet their goal. 34 out of 96 participants (35.4%) included compassionate

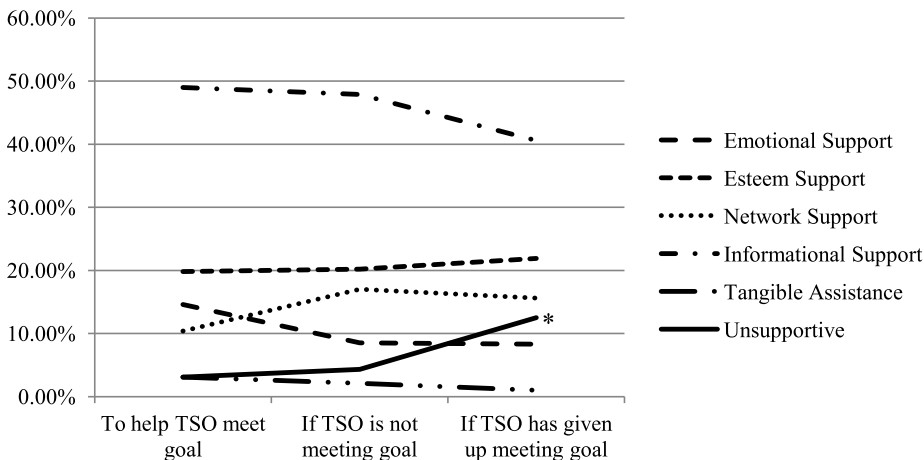

*Significant

**Figure 3 Social support type by messaging scenario.**

content in the messages they generated in the event that the TSO had given up on trying to meet their goal. Cross-tabulations performed for compassionate responses in each of the three messaging scenarios yielded little between-scenario agreement, indicating that compassionate messages were not necessarily written by the same participants across scenarios.

While gain- or loss-framed messages never constituted the majority the messages in a grouping, there were changes in the number of gain- and loss-framed messages by messaging scenario (see Fig. 2). The number of participants who generated gain-framed messages remained stable in the first two messaging scenarios, with 19 out of 96 participants (20.2%) generating gain-framed messages to help their TSO meet their goal and in the event that their TSO was struggling to meet their goal. This percentage dropped to 12 out of 96 participants (12.8%) in the third messaging scenario, in which the TSO had given up on trying to meet their goal. These changes were not significant. The number of participants generating loss-framed messages rose significantly from the first to the third messaging scenario ($Z = -3.55$, $P > .001$, two-tailed). 8 out of 96 participants (8.5%) generated loss-framed messages for their TSOs to help them meet their goal. This number nearly doubled to 15 out of 96 participants (16.1%) in the event that the TSO was struggling to meet their goal, and again to 27 out of 96 participants (28.7%) in the event that the TSO had given up on trying to meet their goal.

None of the changes in social support type were found to be significant across the three messaging scenarios (see Fig. 3), with the exception of a significant spike in the number of messages coded as unsupportive between the second and third scenario ($Z = -2.85$, $P < .01$, two-tailed). We ran a crosstab to determine the relationship between compassion and the various categories within the social support coding schema (e.g., emotional, esteem, network and informational support). Across scenarios, emotional, esteem, and network support messages constituted 85% of all compassionate messages. In the first messaging scenario, informational messages constituted 75% of all messages coded as not

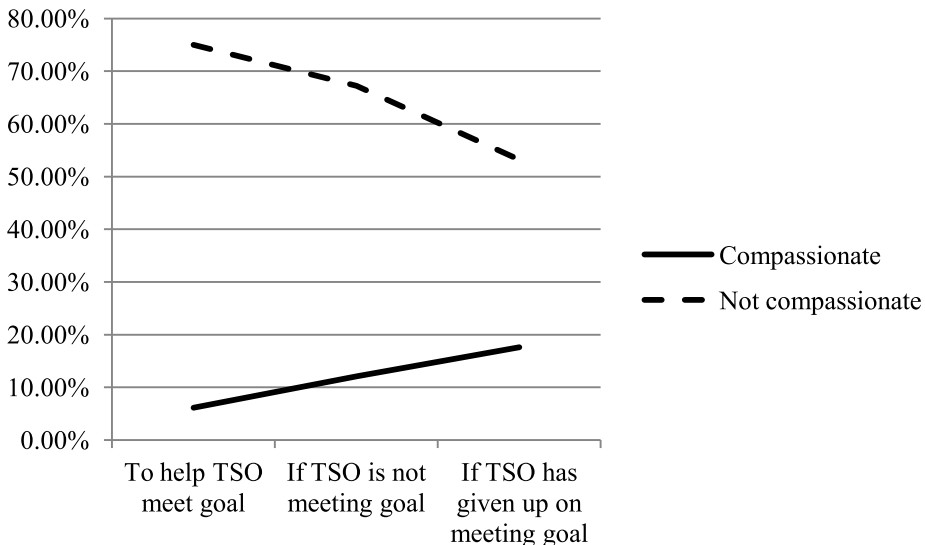

**Figure 4 Compassionate vs. not compassionate messages coded as informational support.**

compassionate, but decreased to 53.2% of all non-compassionate messages by the third messaging scenario (see Fig. 4).

## Message content associations

We looked for significant differences in message content based on both demographic variables and a number of questions we posed to CSOs about their relationships to their respective TSOs, their involvement or lack thereof in the TSOs' efforts to meet their goals, and their perceptions of the TSOs' individual change processes.

While no significant differences were found based on demographic variables, a number of significant differences in message tone and compassion existed based on variables relating to the CSOs' perceptions of their TSOs. CSOs were significantly more likely to generate messages with a positive tone to help their TSOs meet their goals if they felt that their TSOs were highly motivated to achieve that goal, and conversely more likely to generate messages with a negative tone if they felt that their TSOs were unmotivated ($F_{2,90} = 4.14$, $P < .05$). This was also true when generating messages in the event that their TSO was struggling to meet their goal ($F_{2,92} = 6.17$, $P < .01$). This relationship was not significant when the CSOs generated messages in the event that the TSOs had given up on trying to meet their goals. Similarly, CSOs were significantly more likely to generate compassionate messages to help their TSOs meet their goals if they felt that their TSOs were highly motivated to achieve those goals, and less compassionate messages if they felt that their TSOs were unmotivated ($F_{1,91} = 5.64$, $P < .05$), and the same held true in the event that the TSO was struggling to meet their goal ($F_{1,93} = 6.15$, $P < .05$). As was the case with tone, this relationship ceased to be significant when the CSOs generated messages for the scenario in which the TSOs had given up on trying to meet their goals.

CSOs who reported that they would expect a grateful reaction from their TSO were they to offer suggestions or support were significantly more likely to generate messages

with a positive tone to help their TSO meet their goal ($F_{2,90} = 11.61, P < .001$), in the event that the TSO was struggling to meet their goal ($F_{2,92} = 5.51, P < .01$), and in the event that the TSO had given up on trying to meet their goal ($F_{2,92} = 3.58, P < .05$). CSOs who reported an expectation of a grateful reaction to their suggestions or support were also significantly more likely to generate messages with a compassionate tone in all three respective messaging scenarios ($F_{1,91} = 8.85, P < .01; F_{1,93} = 15.91, P < .001; F_{1,94} = 5.29, P < .05$). CSOs who reported that they would expect an annoyed reaction from their TSO to CSO suggestions or support were significantly more likely to generate messages with a negative tone to help their TSO meet their goal ($F_{2,90} = 3.90, P < .05$) and in the event that the TSO was struggling to meet their goal ($F_{2,92} = 6.21, p < .01$). This relationship ceased to be significant in the event that the TSO had given up on trying to meet their goal. CSOs who reported an expectation of an annoyed reaction to their suggestions or support were also more likely to generate non-compassionate messages to help their TSO meet their goal ($F_{1,91} = 9.04, P < .01$) and in the event that the TSO was struggling to meet their goal ($F_{1,93} = 9.20, P < .01$). As with tone, this relationship ceased to be significant in the event that the TSO had given up on trying to meet their goal.

There was a significant relationship between tone and compassion when combined for all messaging scenarios ($r = .527, P < .001$). Multiple linear regression for the first messaging scenario (to help the TSO meet their goal) revealed that tone, rather than compassion, remained significantly positively associated with TSO motivation ($t = 1.99, P < .05$) when both variables were entered as predictors. Similarly, binary logistic regression revealed that only tone remained significantly associated with the expectation of a grateful reaction (Wald $= 11.70, P < .001$). Neither predictor remained significantly associated with the expectation of an annoyed reaction, but the overall model remained significant.

Despite variations between subjects in their relationship to their TSO (e.g., spouse, parent, friend), the type of goal they identified for their TSO (e.g., physical health and well-being vs. competence and mastery), their perception of the severity of consequences the TSO would experience were they to fail to meet their goal, and their degree of frustration with the TSO over his or her target goal or behavior, no significant differences in the tone, compassion, or gain- and loss-framing of messages were found based on these variables.

## DISCUSSION

To our knowledge, this study is the first to qualitatively examine the content of text-based messages generated by CSOs based on the CSO's desire for their TSO to meet a specific goal in a range of behavior change scenarios. It is also the first study to examine the relationship between these messages and CSOs' perceptions of their TSOs and the CSO-TSO relationship. Results of this study indicate that the messages generated by CSOs are significantly more likely to be positive or ambiguous in tone than they are to be negative. Messages are significantly more likely to lack compassionate content than they are to be compassionate regardless of the specific CSO-TSO relationship or the scenario for which each message was generated. Additionally, the presence of compassionate content

in CSO-generated messages was positively associated with three specific subtypes of social support: emotional support, esteem support and network support.

Results further suggest that CSOs' perceptions of TSO motivation and CSOs' expectations of certain positive or negative reactions to suggestions or support are better predictors of the tone and compassion of CSO messages than variables such as CSO frustration or perceived severity of the consequences of the TSO failing to meet the goal. This has important implications for training CSOs on how their expectations of a reaction may drive a negative feedback loop beyond the TSO's actual behavior. Finally, results indicate that while CSOs are more likely to generate messages without gain- or loss-framed content, the number of CSOs who generate loss-framed messages increases as the TSOs struggle to meet or give up on their goals.

## LIWC analysis

The results of a basic linguistic analysis yielded few differences, even when findings were statistically significant between scenarios. However, two findings were interesting. First, sentences were shorter and CSOs used more punctuation in their messages for the third scenario. This could suggest that CSO messages become more emphatic as the behavior change scenario becomes more dire, particularly given that punctuation is often used to convey emotion.

While there were no changes in social and instructive or informational content across scenarios, there was a significant linear decline in the number of positive emotion words that CSOs chose to include in messages from the first to the third scenario, possibly reflecting loss of hope as the TSO gives up on their goals. Interestingly, there was no complimentary increase in the employment of negative emotion words CSOs chose to use. It is possible that, much like compassion, which remained stable across scenarios, this reflects CSO sensitivity toward the TSO in the face of behavior change struggles, even when coupled with a decrease in positivity.

## Motivation

One of the principle findings of this study was that messages were significantly more likely to be positive in tone if the CSO perceived their TSO to be highly motivated to meet their goal, and negative if the CSO perceived their TSO to be unmotivated to meet their goal. These findings suggest that CSOs may be more inclined to be encouraging and affirmative of their TSO when they feel that the TSO is making a concerted effort to change, and more critical or dismissive when they feel that the TSO is not trying hard enough. Although this is, to our knowledge, the first time the persuasive language choices of CSOs have been linked to TSO motivation, these findings resonate with the literature on social reciprocity and its effect on caregiving (*Horowitz & Shindelman, 1983*). It may also be that CSOs are aware that more motivated individuals are more likely to meet their goals, leading CSOs to be more optimistic about their TSOs' chances of succeeding, and therefore more positive in their language choices within motivational messages. The fact that the relationship between CSOs' perceptions of TSO motivation and the tone of CSO messages vanishes in the third messaging scenario is perhaps unsurprising, due to the fact that a

scenario in which TSOs have given up on meeting their goals is by nature one in which TSO motivation is no longer a factor.

Similarly, messages in the first two messaging scenarios were significantly more likely to include compassionate content if the CSO perceived the TSO to be highly motivated to meet their goal. This is, arguably, the more difficult relationship to explain due to the fact that more motivated individuals may have less of a need for expressions of care or concern from their CSOs than those who are less motivated to change. However, these findings may suggest that CSOs are more disposed to be sympathetic to people who they perceive to be trying hard to meet their goal, and that this manifests as compassion within messages. Again, the fact that this relationship does not appear to be significant for the third messaging scenario is likely to be a result of the fact that motivation does not factor into this category of messages.

## Expected reactions to CSO feedback

The tone of messages was also found to vary significantly based on two expected TSO reactions to CSO suggestions or support. CSOs who expected their TSOs to be grateful for feedback were significantly more likely to write positive and compassionate messages, regardless of the messaging scenario. Conversely, CSOs who expected their TSOs to be annoyed upon receiving feedback were significantly more likely to write negative, uncompassionate messages for the first two messaging scenarios.

These findings were of particular interest to investigators, given that so many variables that were expected to influence the tone and compassion of messages (e.g., the perceived closeness of the CSO-TSO relationship or the CSOs' reported frustration with their TSOs' progress toward goal attainment) were not found to be significant predictors. The relationships that were found between the CSO's expectations of their TSO and the message content they generated have important implications. Namely, the CSO's expectation of a positive and affirming reaction to support was related to the generation of positive, compassionate message content. By contrast, the expectation of a negative, irritated reaction to support was related to the generation of negative, uncompassionate message content. As language expectancy theory predicts, these findings suggest that CSOs' expectations of a positive or negative reaction to support has an impact on how they choose to communicate with their TSO—even in non-face-to-face interactions. What's more, they suggest that CSOs' expectations of gratitude or annoyance from their TSOs have more weight in influencing the tone and compassion of their messages than factors such as how well the TSO is doing in trying to meet their goal, how severe the consequences would be if the TSO were to fail, or how frustrated the CSO is with the TSO's progress.

## Supportive communication type

With the addition of a category for unsupportive messages, nearly all messages were successfully categorized based on the social support coding schema developed by *Cutrona & Suhr (1994)*. Although the most common type of supportive communication across scenarios was informational support, an equivalent number of messages fell into the emotional, esteem, or network support categories for each scenario, all of which include

expressions of connectedness, love, care and liking. Very few tangible assistance messages were generated, which was to be expected given that we asked CSOs to generate brief text-based messages that were intended to be a form of stand-alone support. Unsurprisingly, the number of explicitly unsupportive messages spiked in the scenario in which CSOs had given up on their goals completely, although even in this third scenario the number of unsupportive messages was relatively low. This may be tied to the relationship that appears to exist between negative tone in messages and CSO perceptions of low TSO motivation. Namely, CSOs may be more likely to withdraw support completely if they feel that their TSO is making no effort to meet their goal.

### Gain- and loss-framing

Gain- and loss-framed messages, which stressed the potential positive outcomes of goal achievement or the potential negative consequences of goal failure, respectively, constituted a significant portion of the messages in all messaging scenarios. Most notably, there was a significant increase in the number of loss-framed messages generated by CSOs from the first to the third messaging scenario. As the scenarios became more indicative of TSOs' failure to meet their goals, CSOs became more disposed to stress the potential negative consequences of these failures within their motivational messages. This corresponds with the decrease in positive affect words from the first to the third messaging scenario as identified in the LIWC analysis. Although this increase in loss-framed messages by scenario is not unexpected, future research should examine whether this is a desirable trend in communication between CSOs and TSOs about behavior change. Recent research indicates that recipients of motivational messages generally prefer to receive gain-framed over loss-framed content but may prefer loss-framed messages when motivation is lower (*Muench et al., 2014*; *Muench et al., 2013*). This tendency may generalize to all communication patterns—even those generated by CSOs.

### Limitations

While the scope of this line of research was exploratory, there were a number of limitations. Most notably, the representativeness of this CSO sample was limited by the age of participants and because nearly three quarters of the participants were male. Some important and common CSO-TSO relationships, including CSO parent—TSO child relationships, could not be assessed based on the demographic characteristics of the sample, limiting the generalizability of the findings to the spectrum of CSO-TSO relationships. Furthermore, we did not restrict the availability of the survey only to CSOs of TSOs who might be the target of a true health intervention, but allowed CSOs to generate a wide range of goals, many of which were not explicitly health related. It is possible that this broader group of CSOs may generate different message content than a CSO sample with specific health or mental health-related goals for their TSOs. However, it bears repeating that we found no differences in tone or compassion based on goal type.

Another limitation is that we did not ask CSOs to generate brief text-based messages in a naturalistic context (e.g., via text message on a mobile phone or in an online one-on-one chat setting), but rather in an online survey setting in which there was no risk of the TSO

receiving the messages they generated. It is possible that asking CSOs to generate messages via mobile text message or web chat, or in a setting in which their TSO could see the message, would yield different results. Nevertheless, our results correspond with much of the existing literature on CSOs, behavior change and language expectancy, indicating that further research on CSO-generated messages may be helpful to our understanding of CSOs' persuasive tactics.

While message coding was standardized, results of our analyses of tone, compassion and gain- and loss-framing should be interpreted with reservation as these coding categories were operationalized by researchers rather than using an external standard. In addition, understanding the nuances of the coding categories is important. For example, a message that contained only a direction or instruction (e.g., "Stop smoking now.") was not coded as compassionate, due to the fact that such a message does not explicitly convey a tone of care or concern. However, although the message was therefore coded as uncompassionate, it should not necessarily be interpreted as antonymous with compassion (e.g., cruel or indifferent). Future research should aim to refine these coding schemas to improve our understanding of subtle variations in message content.

Despite these limitations, we believe that the findings of this study hold promise in opening a line of research about how CSOs use language to motivate TSOs and how to improve communication between these populations to improve treatments that target the CSO-TSO relationship.

## Conclusions and future research

When taken together, these findings underscore the importance of attending to patterns in the language choices of CSOs when addressing their TSOs about goal achievement or failure, and how certain variables in the CSOs' perceptions of their TSOs and their interpersonal dynamics moderate these characteristics. As we work toward a goal of fostering healthier relationships that promote change in people's daily lives beyond the clinic, our recommendation for future research is to prioritize elucidating the underlying communication dynamics between CSOs and TSOs, especially in text-based communication. These efforts may improve our ability to target counterproductive or ineffective communication patterns while providing tools and resources that enhance productive and supportive communication between these two populations.

### Funding

This research was supported with funding from the National Institute on Alcohol Abuse and Alcoholism (R34 AA021502-01). The funders had no role in study design, data collection and analysis, decision to publish, or preparation of the manuscript.

### Grant Disclosures

The following grant information was disclosed by the authors:
National Institute on Alcohol Abuse and Alcoholism: R34 AA021502-01.

## Competing Interests

Frederick Muench is an Academic Editor for PeerJ.

## Author Contributions

- Katherine van Stolk-Cooke conceived and designed the experiments, performed the experiments, analyzed the data, wrote the paper, prepared figures and/or tables, reviewed drafts of the paper.
- Marie Hayes analyzed the data, reviewed drafts of the paper.
- Amit Baumel reviewed drafts of the paper.
- Frederick Muench conceived and designed the experiments, performed the experiments, analyzed the data, contributed reagents/materials/analysis tools, wrote the paper, reviewed drafts of the paper.

## Human Ethics

The following information was supplied relating to ethical approvals (i.e., approving body and any reference numbers):

New York State Psychiatric Institute Institutional Review Board, IRB approval 6625.

## Supplemental Information

Supplemental information for this article can be found online at http://dx.doi.org/10.7717/peerj.1151#supplemental-information.

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
