# Peer review of "Understanding text-based persuasion and support tactics of concerned significant others"

_PeerJ, doi:10.7717/peerj.1151_

## Round 0.1 · original submission · Minor Revisions

· Academic Editor

Minor Revisions

Dear Authors,Please revised the manuscript accordingly and resubmit it as soon as possible as the editing required seems minor. Do read carefully the revisions required especially suggested by the first Peer Reviewer.

·

Basic reporting

This paper is well written and very interesting. Minor comments include:
- First sentence of the introduction states an increased focus on CSO’s but the source is form 1989- might be good to have a more recent source for this information.
- Would be good to say whether programs like CRAFT were successful in getting people into treatment (for example gambling research involving the CRAFT program found it was not in increasing treatment uptake) (line 54-57)
- Be great to substantiate the statement of ‘more honest’ (line 63)
- Lines 65-66 would be good to be more specific and refer to mobile technology rather than technology based interventions. This would help the transition to the next lines re communication and messages although the jump back to CSO’s is a leap.
- The introduction at time jumps between terminology – CMC, technology, mobile,
- Adams, Baumer and Gay, 2014 not on reference list

Experimental design

Could the aim include reference to the specific variables so that the results and discussion are easier to follow? (e.g., meet goal, not meet goal, given up) and define ‘certain variables’ (i.e., motivation etc.) and message characteristics (e.g., tone etc)

Validity of the findings

- The opening sentence of the discussion is difficult to follow due to its length and volume of information. It is quite difficult to link this back to the aims of the study.
- Lines 407 to 409 do not currently add to interpreation of the results. It would be better if the literature described was more specific to the current findings and also provides a source of the information.
- Line 416 introduces “informational processes” which is a new term meant to convey the types of support provided. Would be good to be consistent throughout.
- I’m not sure about Lines 431-434 and the link to therapeutic alliance and treatment seeking. It might be better to compare these findings to CSO interventions identified in the intro such as CRAFT and whether perceived motivation was a part of this type of program.
- Difficult to follow lines 459 to 463 due to its length and number of different ideas.

Comments for the author

The authors are to be congratulated on their interesting and informative work that explores an overlooked area of communication.

Reviewer 2 ·

Basic reporting

This study that examined the language content of brief text-message generated by significant others to motivate a person to achieve behavioral goals is reported in a clear, comprehensive, and 'self-contained' manner.

Experimental design

The study design is appropriate and well-justified.

Validity of the findings

The findings appear valid. The results are presented very clearly and the general limitations are well-described.

Comments for the author

Generally, this report is worth published.

---

## Round 0.2 · accepted · Accept

· Academic Editor

Accept

Dear Authors,

Thank you for your revised manuscript.It has been re-reviewed .The peer reviewers have found the revisions made to be satisfactory thus the manuscript has been accepted for further processing. Thank you again for your submission.

·

Basic reporting

Submission meets standards outlined.

Experimental design

Submission meets standards outlined

Validity of the findings

Data are well presented and clear

Comments for the author

I am satisfied that the authors have responded to feedback provided. The article is very interesting, well presented and presents an innovative and thoughtful research design.